# Feedback perceptions of first year medical residents: An intervention-based survey study

Rachel Cox◎*⊕, John Arthur⊕, Kathryn Burtson

Department of Internal Medicine, Wright Patterson AFB and Wright State University, Wright-Patterson AFB, Ohio, United States of America

⊕ These authors contributed equally to this work.
* rachel.s.cox5.mil@health.mil

## Abstract

### Background

Feedback in residency is a necessity for progression toward clinical competency and is included in The Accreditation Council for Graduate Medical Education (ACGME) milestones as an essential component for accreditation.

### Purpose

Our study elucidates perceptions of feedback of first-year residents and aims to identify how these perceptions change after education on building expertise through deliberate practice.

### Methods

First-year internal medicine and neurology residents of a mid-sized university-affiliated residency program answered a five-question 5-point unipolar response scale questionnaire regarding feedback perceptions before and after attending a workshop about building expertise through effective feedback during residency orientation. Related-Samples Wilcoxon Signed Rank Test was applied for comparing pre- versus post-questionnaire data.

### Results

Of 31 first-year residents, 29 completed the pre-questionnaire for a completion rate of 93.5%, while 24 of 31 completed the post-questionnaire for a completion rate of 77.4%. Of the five questions, three improved when comparing pre and post responses to, including the questions on confidence in the ability to procure feedback (p = <0.001), the effort put into procuring feedback (p = 0.001), and frequency of seeking feedback (p = 0.002). Interest in receiving feedback and the importance of feedback remained unchanged after workshop attendance.

### Conclusion

Residents should be educated on building expertise through deliberate practice and how to obtain high-quality feedback, given the emphasis and essentiality of feedback within the milestone assessment system and the core competencies of ACGME. In our study,

**Data Availability Statement:** All relevant data are within the manuscript and its Supporting Information files.

**Funding:** The author(s) received no specific funding for this work.

**Competing interests:** The authors have declared that no competing interests exist.

education on these topics led to significant improvement in resident perceptions of confidence in the ability to procure feedback, effort put into procuring feedback, and frequency at which feedback would be sought.

## Introduction

In 2013, the ACGME incorporated milestones in trainee assessment as a critical element in the Next Accreditation System. Per "The Milestones Guidebook" by the ACGME in 2020, feedback is an essential and required activity for achieving each milestone within the Six Core Competencies as it is believed feedback is a necessity to achieve clinical competence [1, 2]. It is clear that among residency programs, faculty and residents understand just how important feedback is and most residents actively try to change their behavior or practices based on the feedback received [3–5].

Despite this well-known importance, obtaining quality feedback among resident trainees remains variable. Although residents endorse a growth mindset orientation, one survey revealed residents found the act of seeking feedback to induce anxiety and led to subtle tension between supervisors and trainees [6]. In one large study of 17 emergency medicine programs, attending physicians initiated most of the feedback rather than the trainees [7]. Attending physician-initiated feedback is suboptimal as feedback is a shared responsibility. Trainees should first set their own goals prior to obtaining feedback as unstructured feedback is general and not specific while regular, specific feedback promotes trainee empowerment [8]. Feedback frameworks such as the ACGME's Ask-Discuss-Ask-Plan Together model allows for a shared educational alliance that reduces anxiety and promotes continued growth in both the learner and physician [9].

Focused, specific goal setting by trainees and frequent trainee initiated feedback promotes constant progress toward expertise level without falling into automaticity, a concept coined deliberate practice, wherein trainees mindfully push past their comfort zone to achieve improvement [10]. Deliberate practice was superior to traditional clinical medical education in achieving specific clinical skill acquisition goals; however, deliberate practice requires trainees to identify their weaknesses, something that health professionals have difficulty doing [11–13].

Residents are not empowered to seek feedback as they are often not educated on how to seek regular and specific feedback within their training. There is no current formal or standardized training on resident feedback despite incorporation of feedback into the ACGME required milestones. Education on how to seek quality feedback can improve deliberate practice, feedback behaviors, and professional growth and development.

## Materials and methods

This study was conducted at Wright State University Department of Medicine in a single day during the first-year Wright State resident orientation. Participants consisted of thirty-one incoming 1st-year internal medicine and neurology residents who were required to attend a 90-minute workshop about building expertise through effective feedback, presented by the Wright State Internal Medicine Associate Program Director. Before the workshop, residents anonymously answered a five-question 5-point unipolar response scale questionnaire that would determine their confidence in the ability to procure feedback, interest in receiving feedback, effort in procuring feedback, the importance of feedback, and frequency of seeking feedback. The questionnaire including the 5-point unipolar response scale was developed using the AMEE guide for educational research, with responses within the scale represented nominally

as such: 1 = never, 2 = once in a while, 3 = sometimes, 4 = often, and 5 = almost always. The questionnaire was sent via a SurveyMonkey link to their mobile devices, where they created their unique username. All survey materials were developed and distributed by the Wright State Internal Medicine Program Director.

The content of the workshop focused on deliberate practice with its components consisting of the following: having specific goals, individualized practice, being outside your comfort zone, and continuous feedback with associated modifications to one's behaviors as needed. Furthermore, residents were taught to develop specific, measurable, attainable, realistic, and timely goals. The core learning strategy utilized was the think-pair-share concept to assist residents in formulating their own individualized goals and assessing them for quality allowing for more effective feedback by faculty and staff. This workshop was developed using Kern's 6-step model, including (i) problem identification and general needs assessment, (ii) targeted needs assessment, (iii) goals and objectives, (iv) educational strategies, (v) implementation, and (vi) feedback and evaluation.

Upon completion of the workshop, participants would complete the same questionnaire using their unique username, to determine if there were any changes in their perceptions of feedback. After the conversion of nominal values to scale values, a Related-Samples Wilcoxon Signed Rank Test was applied to compare the pre- and post-questionnaire data. This activity was reviewed by the Wright-Patterson Medical Center Human Research Protection Program (Protocol #: FWP20230001N), and determined to not constitute research following Federal Regulations and Department of Defense Instructions.

## Results

Of 31 first-year residents, 29 completed the pre-questionnaire for a completion rate of 93.5%, while 24 of 31 completed the post-questionnaire for a completion rate of 77.4%. Table 1 reveals that on pre to post-questionnaire comparison, using the Related-Samples Wilcoxon Signed Rank Test as described above, three of the five questions show improvement on analysis including the questions regarding confidence in ability to procure feedback with improvement from $3.52\pm0.57$ to $4.37\pm0.58$ ($p<0.001$, ES = 0.59), effort put into procuring feedback with improvement from $4.14\pm0.74$ to $4.50\pm0.59$ ($p = 0.001$, ES = 0.26), and how often feedback is sought from a faculty member with improvement from $4.34\pm0.55$ to $4.75\pm0.53$ ($p = 0.002$, ES = 0.35). In contrast, the questions regarding interest in receiving feedback and the importance of feedback did not change at $4.83\pm0.38$ to $4.83\pm0.48$ ($p<0.56$) and $4.79\pm0.49$ to $4.87\pm0.45$ ($p = 0.08$) respectively.

## Discussion

Residents are not empowered to seek feedback as they are often not taught how to effectively seek quality feedback in residency. Education on feedback may lead to improvement in feedback behaviors, deliberate practice, and professional growth and development.

**Table 1. Analyses of pre and post feedback perceptions questionnaire responses.**

| | Pre (n = 29) | SD | Post (n = 24) | SD | P value | Effect size |
|---|---|---|---|---|---|---|
| How confident are you in your ability to procure feedback in your residency training? | 3.52 | 0.57 | 4.37 | 0.58 | <0.001 | 0.59 |
| How interested are you in receiving feedback throughout your residency training? | 4.83 | 0.38 | 4.83 | 0.48 | <0.56 | N/A |
| How much effort will you put into procuring feedback during residency training? | 4.14 | 0.74 | 4.50 | 0.59 | 0.001 | 0.26 |
| How important is feedback to your residency training? | 4.79 | 0.49 | 4.87 | 0.45 | 0.08 | N/A |
| How often will you seek feedback from a faculty member in your residency training? | 4.34 | 0.55 | 4.75 | 0.53 | 0.002 | 0.35 |

Resident perception of confidence in ability to procure feedback improved significantly after education on goal creation and deliberate practice with moderate effect size. The content of the workshop contributes to this relatively larger effect size as residents were educated specifically on how to effectively seek feedback which likely lead to a relatively greater increase in confidence. Although resident confidence in the setting of feedback has not been examined prior to this study, improvement in confidence may lead to less perceived tension and anxiety surrounding obtaining feedback which are known barriers to residents seeking feedback [6].

Resident effort put forth into obtaining feedback also improved significantly with small effect size. Resident effort as it relates to obtaining feedback has not been investigated prior to this study; however, it has been shown that feedback is usually initiated by attending physicians [7]. Through perceived improvement in effort, residents may feel more empowered to seek feedback after education on feedback and deliberate practice.

Resident perception of frequency of seeking feedback improved significantly with small effect size. In previous investigations, regular, specific feedback led to trainee empowerment and perception of seeking feedback more frequently could promote empowerment and advancement toward expertise rather than suboptimal, general feedback [8].

Perceptions of interest in receiving feedback and importance of feedback showed no change as pre-questionnaire ratings of these variables were high before the workshop and remained high on the post-questionnaire signaling that residents are highly interested and value feedback regardless of formal instruction on the topic.

This study's limitations include that it only examines a single mid-sized university-affiliated residency program, only a single year of residents, and only residents within two specialties which limits the ability to generalize the results Although statistical significance was achieved, the small sample size of this study is a limitation. The results of this study may have been altered by the inclusion of non-first year residents, residents across different disciplines, residents within various sized residency programs, or by inclusion of a greater number of residents in general. The ability to replicate this study is unknown at this time.

Additionally, various trainee characteristics including gender, ethnicity, cultural background, individual learning style, previous experiences and medical school background are not known which is a major limitation and could introduce bias as these characteristics could influence their perception of feedback. Although this study was not designed, approved, or powered to assess the effect of these characteristics on feedback, this would be an important area to include in future studies.

Lastly, this quantitative study focuses on level one of the Kirkpatrick Education Model, meaning it examines the participant's reaction to the training. For this reason, it is unknown if residents changed their feedback practices or if this intervention had any contribution to trainee professional development. It is not known whether this intervention lead to any lasting change in perception of feedback or had impact on any other areas of graduate medical education as it pertains to resident growth. Similarly, this study does not investigate specific feedback related outcomes of the workshop to include quality, timeliness, or frequency of feedback sought by trainees.

Future directions of this study could involve examining residents across different years and specialties and could be extended to additional graduate medical education and undergraduate medical education programs. Qualitative methods, such as semi-structured interviews, could elucidate how feedback behaviors were affected by this workshop. Level three of the Kirkpatrick Education Model could also be examined through the lens of this study, which would include resident change in behavior in feedback practices.

## Conclusions

Education on feedback enhances resident perception of confidence, effort, and frequency as it relates to seeking and obtaining feedback in residency and this could lead to higher efficacy and efficiency in reaching ACGME milestones. Residents may feel more empowered to seek feedback after instruction on how to do so effectively. Instruction on feedback as a necessity and standardization by ACGME of this instruction could be beneficial to residency programs nationwide.

## Supporting information

**S1 Dataset. Underlying dataset for pre and post feedback perceptions responses.**
(XLSX)

**S1 Checklist. Human participants research checklist.**
(DOCX)

## Acknowledgments

We thank Dr. Ronald J. Markert, former vice chair for research in Wright State University Department of Internal Medicine, for statistical analysis and content review of the survey data.

## Author Contributions

**Conceptualization:** Rachel Cox, John Arthur, Kathryn Burtson.

**Data curation:** Rachel Cox.

**Investigation:** Kathryn Burtson.

**Project administration:** Kathryn Burtson.

**Resources:** Kathryn Burtson.

**Supervision:** Kathryn Burtson.

**Visualization:** Rachel Cox, John Arthur.

**Writing – original draft:** Rachel Cox, John Arthur, Kathryn Burtson.

**Writing – review & editing:** Rachel Cox, John Arthur, Kathryn Burtson.

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
