## [Decision Letter · Decision Letter 0]

7 Nov 2023

PONE-D-23-29943Feedback perceptions of first year medical residents: A survey studyPLOS ONE

Dear Dr. Cox,

Thank you for submitting your manuscript to PLOS ONE. After careful consideration, we feel that it has merit but does not fully meet PLOS ONE’s publication criteria as it currently stands. Therefore, we invite you to submit a revised version of the manuscript that addresses the points raised during the review process.

Thank you for submitting the article to PLOS One. Your paper has received significant positive feedback from the reviewers. However, there are some areas that the reviewers recommend some revisions and we were ups be happy to consider the article for publication after the review points are addressed. We look forward to hearing from you.

We look forward to receiving your revised manuscript.

Kind regards,

Souparno Mitra, M.D.

Academic Editor

PLOS ONE

Reviewers' comments:

Reviewer's Responses to Questions

**Comments to the Author**

1. Is the manuscript technically sound, and do the data support the conclusions?

Reviewer #1: Yes

Reviewer #2: Yes

Reviewer #3: Yes

Reviewer #4: Yes

2. Has the statistical analysis been performed appropriately and rigorously? 

Reviewer #1: Yes

Reviewer #2: Yes

Reviewer #3: I Don't Know

Reviewer #4: Yes

3. Have the authors made all data underlying the findings in their manuscript fully available?

Reviewer #1: Yes

Reviewer #2: Yes

Reviewer #3: Yes

Reviewer #4: Yes

4. Is the manuscript presented in an intelligible fashion and written in standard English?

Reviewer #1: Yes

Reviewer #2: Yes

Reviewer #3: Yes

Reviewer #4: Yes

5. Review Comments to the Author

Reviewer #1: It's a good study looking into the education of trainees, which is often overlooked and has limited interest among researchers. I like the methodology and the results are suggestive to improving the feedback process for trainees. In terms of the results, I would like to know more about the trainees who took the survey. For example, males/ females, IMG/ AMG, ethnicity, etc. Trainees from varying demographics could have a different perception to the feedback process. If this data is not available, it could be amajor limitation.

Reviewer #2: Title: It is not a simple survey study. This is an intervention-based survey study. Title could be modified a bit to reflect that

Materials and Methods: Line 97- It would be helpful for the readers to know how you scored the 5-point likert scale. Especially, how you scored that the last question on how frequently they will seek feedback. Maybe include this scale as a separate figure.

Discussion: line 139-143- the relatively bigger effect size of confidence in ability to procure feedback compared to other questions in the survey could be explained by the content of the workshop which was geared specifically to train residents to seek feedback more effectively. This could be discussed more here.

Reviewer #3: Excellent study.

1. Authors may consider condensing the length of the abstract as it is somewhat lengthy. Moreover, it is recommended to structure the abstract by providing subheadings for improved clarity. This can be done by organizing the abstract into sections such as Background, Method, Results, and Conclusion.

2. If the trainees involved in the study come from similar backgrounds or have similar perspectives, this can introduce bias into the research or survey study. How this was addressed in this study?

3. There are several potential biases and issues that could arise when using Kern's 6-step model in a research or survey study conducted by trainees. Trainees may have limited experience of survey studies, which can lead to issues with study design, data collection. Was this addresses or discussed in limitations?

4. This study does not explore the factors that may influence residents' perception of feedback, such as their previous experiences, individual learning styles, or cultural background. Understanding these factors could provide a more comprehensive understanding of the limitations and potential biases in the residents' perceptions.

5. The study only focuses on residents' perceptions of feedback and does not evaluate the effectiveness or impact of the feedback on their learning and professional development. It would be valuable to investigate whether the feedback provided to residents actually leads to meaningful changes in their knowledge, skills, and attitudes.

6. The study only considers one aspect of the feedback process, specifically the residents' reactions to the feedback. It does not explore other important dimensions, such as the quality and timeliness of the feedback, and the follow-up actions taken based on the feedback. Investigating these aspects could provide a more comprehensive understanding of the feedback process in residency programs.

7. Lastly, this study only focuses on a single year of residents in a single residency program. Generalizing the findings to other programs or different cohorts of residents should be done with caution.

These limitations should be incoperated within discussion section and discussed in detail.

Reviewer #4: Thanks for giving me the opportunity to review this article.

LIMITATIONS: The sample size is very small. Also like the author has mentioned that perhaps the results would have been different if the second and third year residents were included too. It would have been helpful to include other disciplines besides just internal medicine and Neurology. Also it would be great to see the impact of size of the program ie small sixed ( more cohesive) or big residency program on results of the study.

SOLUTION: Please mention in limitation that perhaps the results could have been different if the sample size was bigger and second and third year residents were included too. The author did touch on this topic briefly in the limitation section. Also it remains to be seen if these results will replicate in different sized residency programs and across different disciplines.

6. PLOS authors have the option to publish the peer review history of their article (what does this mean?). If published, this will include your full peer review and any attached files.

Reviewer #1: **Yes: **Ankit Parmar

Reviewer #2: No

Reviewer #3: No

Reviewer #4: **Yes: **Jasleen Kaur

---

## [Author Response · Author response to Decision Letter 0]

8 Jan 2024

Please find Revised Manuscript with Tracked Changes, original Manuscript, Response to Reviewers, and S1 Dataset which have been uploaded. We appreciate your time and allowing us to revise our manuscript as well as make other corrections including ensuring adherence to the formatting guidelines, making our data available, and reviewing the accuracy of our reference list. We are willing to make any further changes that are needed and we look forward to continuing our publication process with PLOS ONE.

---

## [Decision Letter · Decision Letter 1]

18 Jan 2024

PONE-D-23-29943R1Feedback perceptions of first year medical residents: A survey studyPLOS ONE

Dear Dr. Cox,

Thank you for submitting your manuscript to PLOS ONE. After careful consideration, we feel that it has merit but does not fully meet PLOS ONE’s publication criteria as it currently stands. Therefore, we invite you to submit a revised version of the manuscript that addresses the points raised during the review process.

**ACADEMIC EDITOR: **

Thank you for submitting your manuscript. There is a unanimous decision among reviewers that some minor revisions that would be beneficial for the paper. I encourage the authors to review the points including but not limited to the length of the abstract, the confidence interval and the factors influencing the results. Once the updates are incorporated we would be happy to reconsider the manuscript for further review. 

We look forward to receiving your revised manuscript.

Kind regards,

Souparno Mitra, M.D.

Academic Editor

PLOS ONE

Journal Requirements:

Reviewers' comments:

Reviewer's Responses to Questions

**Comments to the Author**

1. If the authors have adequately addressed your comments raised in a previous round of review and you feel that this manuscript is now acceptable for publication, you may indicate that here to bypass the “Comments to the Author” section, enter your conflict of interest statement in the “Confidential to Editor” section, and submit your "Accept" recommendation.

Reviewer #5: All comments have been addressed

Reviewer #6: All comments have been addressed

Reviewer #7: All comments have been addressed

2. Is the manuscript technically sound, and do the data support the conclusions?

Reviewer #5: Yes

Reviewer #6: Yes

Reviewer #7: Yes

3. Has the statistical analysis been performed appropriately and rigorously? 

Reviewer #5: N/A

Reviewer #6: Yes

Reviewer #7: I Don't Know

4. Have the authors made all data underlying the findings in their manuscript fully available?

Reviewer #5: Yes

Reviewer #6: (No Response)

Reviewer #7: Yes

5. Is the manuscript presented in an intelligible fashion and written in standard English?

Reviewer #5: Yes

Reviewer #6: Yes

Reviewer #7: Yes

6. Review Comments to the Author

Reviewer #5: The authors have responded to all the reviewers comments and also formatted and adjusted the manuscript. The authors' responsiveness reflects their commitment to producing a high-quality piece of work. With these commendable revisions, I believe the manuscript is now well-prepared for the next stage of the review process.

Reviewer #6: Dr. Souparno Mitra, M.D.

PLOS ONE

I have had the honor of reviewing the revised version article titled "Feedback perceptions of first year medical residents: An intervention-based survey study" submitted to PLOS ONE for peer review.

The manuscript addresses a very important as well as a critical aspect of medical education that is feedback in residency, which is a core element for clinical competency. The study aligns with ACGME milestones and contributes to the understanding of residents' perceptions and behaviors related to feedback.

The study results have few limitations as identified by the authors, mainly the study focuses on a single mid-sized university-affiliated residency program, a single year of residents, and only two specialties. The sample size is very small with N=24 who completed a post questionnaire. The findings may not be fully generalizable to different specialties, residency programs, or years. The study primarily focuses on level one of the Kirkpatrick Education Model, examining participants' reactions to training. Future research could explore the impact on residents' actual feedback practices (level three) using qualitative methods. To enhance generalizability, consider including residents from different specialties, years, institutions and different demographics. This could provide a broader understanding of how feedback perceptions vary across diverse settings. It looks like all the above-mentioned limitations were raised by the reviewers and the authors made diligent efforts to disclose those in their revised manuscript.

Reviewer #7: Seems like an interesting study, wonder if there will be longitudinal follow up to ascertain improvement in seeking feedback.

Please see attached file

7. PLOS authors have the option to publish the peer review history of their article (what does this mean?). If published, this will include your full peer review and any attached files.

Reviewer #5: **Yes: **Aditi Sharma

Reviewer #6: **Yes: **Surya Karlapati MD

Reviewer #7: No

---

## [Author Response · Author response to Decision Letter 1]

11 Feb 2024

Reviewer(s)’ Comments to Author: 

Reviewer: Academic Editor 

Thank you for submitting your manuscript. There is a unanimous decision among reviewers that some minor revisions that would be beneficial for the paper. I encourage the authors to review the points including but not limited to the length of the abstract, the confidence interval and the factors influencing the results. 

Thank you for the feedback. The length of the abstract is currently 259 words decreased from 288 in the original version with a known limitation of 300 words per PLOS formatting guidelines. The confidence intervals were generated using Related-Samples Wilcoxon Signed Rank Test via SPSS. The various factors influencing the results and consequent limitations of these factors have been included in the discussion section. 

Reviewer: 5 

The authors have responded to all the reviewers comments and also formatted and adjusted the manuscript. The authors' responsiveness reflects their commitment to producing a high-quality piece of work. With these commendable revisions, I believe the manuscript is now well-prepared for the next stage of the review process. We agree that this is not a simple survey study, the title was altered to reflect that. 

We appreciate the feedback. 

Reviewer: 6 

I have had the honor of reviewing the revised version article titled "Feedback perceptions of first year medical residents: An intervention-based survey study" submitted to PLOS ONE for peer review.

The manuscript addresses a very important as well as a critical aspect of medical education that is feedback in residency, which is a core element for clinical competency. The study aligns with ACGME milestones and contributes to the understanding of residents' perceptions and behaviors related to feedback.

The study results have few limitations as identified by the authors, mainly the study focuses on a single mid-sized university-affiliated residency program, a single year of residents, and only two specialties. The sample size is very small with N=24 who completed a post questionnaire. The findings may not be fully generalizable to different specialties, residency programs, or years. The study primarily focuses on level one of the Kirkpatrick Education Model, examining participants' reactions to training. Future research could explore the impact on residents' actual feedback practices (level three) using qualitative methods. To enhance generalizability, consider including residents from different specialties, years, institutions and different demographics. This could provide a broader understanding of how feedback perceptions vary across diverse settings. It looks like all the above-mentioned limitations were raised by the reviewers and the authors made diligent efforts to disclose those in their revised manuscript.

We appreciate the opportunity to highlight further limitations and direction of future study. 

Reviewer: 7

Seems like an interesting study, wonder if there will be longitudinal follow up to ascertain improvement in seeking feedback.

Please see attached file

A citation was added to highlighted sentence in line 73 as noted in the comment in attached file by Reviewer 7, we appreciate the feedback. We recognize that there is no longitudinal component to this study as noted in the discussion, this is certainly an area for study in the future for follow up on these findings.

---

## [Decision Letter · Decision Letter 2]

23 Feb 2024

Feedback perceptions of first year medical residents: An intervention-based survey study

PONE-D-23-29943R2

Dear Dr. Cox,

We’re pleased to inform you that your manuscript has been judged scientifically suitable for publication and will be formally accepted for publication once it meets all outstanding technical requirements.

Kind regards,

Souparno Mitra, M.D.

Academic Editor

PLOS ONE

Additional Editor Comments (optional):

Reviewers' comments:

Reviewer's Responses to Questions

**Comments to the Author**

1. If the authors have adequately addressed your comments raised in a previous round of review and you feel that this manuscript is now acceptable for publication, you may indicate that here to bypass the “Comments to the Author” section, enter your conflict of interest statement in the “Confidential to Editor” section, and submit your "Accept" recommendation.

Reviewer #2: All comments have been addressed

Reviewer #3: All comments have been addressed

Reviewer #8: All comments have been addressed

2. Is the manuscript technically sound, and do the data support the conclusions?

Reviewer #2: Yes

Reviewer #3: Yes

Reviewer #8: Yes

3. Has the statistical analysis been performed appropriately and rigorously? 

Reviewer #2: Yes

Reviewer #3: I Don't Know

Reviewer #8: I Don't Know

4. Have the authors made all data underlying the findings in their manuscript fully available?

Reviewer #2: Yes

Reviewer #3: Yes

Reviewer #8: Yes

5. Is the manuscript presented in an intelligible fashion and written in standard English?

Reviewer #2: Yes

Reviewer #3: Yes

Reviewer #8: Yes

6. Review Comments to the Author

Reviewer #2: The authors seem to have made all recommended edits and the article now appears ready for publication. This study addresses the important topic of first year residents' perception for receiving effective feedback and discussed how this perception changes with formal training in receiving feedback.

Reviewer #3: The authors have successfully addressed all questions and feedback provided, demonstrating thorough comprehension and effective integration of suggestions into their work.

Reviewer #8: All the comments by the reviewer are addressed in this revised manuscript. Revised script seems much better than the original manuscript.

7. PLOS authors have the option to publish the peer review history of their article (what does this mean?). If published, this will include your full peer review and any attached files.

Reviewer #2: No

Reviewer #3: No

Reviewer #8: No

---

## [Editor Report · Acceptance letter]

28 Mar 2024

PONE-D-23-29943R2 

PLOS ONE

Dear Dr. Cox, 

I'm pleased to inform you that your manuscript has been deemed suitable for publication in PLOS ONE. Congratulations! Your manuscript is now being handed over to our production team.

Kind regards, 

on behalf of

Dr. Souparno Mitra 

Academic Editor

PLOS ONE